# Echocardiographic Features of the Ductus Arteriosus and the Foramen Ovale in a Hospital-Based Population of Neonatal Foals

**DOI:** 10.3390/ani12172242

**Published:** 2022-08-30

**Authors:** Lisa De Lange, Ingrid Vernemmen, Gunther van Loon, Annelies Decloedt

**Affiliations:** Equine Cardioteam, Department of Internal Medicine, Reproduction and Population Medicine, Faculty of Veterinary Medicine, Ghent University, Salisburylaan 133, 9820 Merelbeke, Belgium

**Keywords:** equine, cardiology, neonatology

## Abstract

**Simple Summary:**

In fetal circulation, the distribution of oxygenated blood from the maternal placenta is facilitated by two intracardiac shunts. Oxygenated blood flows from the right atrium to the left heart through the foramen ovale, which is formed by the septum primum and septum secundum. Less oxygenated blood is directed to the placenta through the ductus arteriosus, which connects the pulmonary artery and the aorta. The ductus arteriosus and foramen ovale should close after birth. However, knowledge about the exact time of closure of those structures in foals is limited. The current study investigates the ultrasonographical closure of both the ductus arteriosus and the foramen ovale in healthy and diseased neonatal foals. Cardiac auscultation and ultrasound were performed on fifty foals. Cardiac murmurs were common, and in some foals, the ductus arteriosus was still open at ten days of age. The foramen ovale was not open; however, a fluttering motion of the septum primum into the left atrium was a common finding in healthy and diseased foals. The exact clinical importance of those findings needs to be further elucidated.

**Abstract:**

The ductus arteriosus (DA) and foramen ovale (FO), including the septum primum (SP) and septum secundum (SS), are important structures in fetal circulation and are unexplored in neonatal equids. The objective of this study is to describe echocardiographic characteristics in a hospital-based population of neonatal foals. On days 2, 5 and 10 after parturition, cardiac ultrasound was performed, and clinical data were collected in healthy and diseased Warmblood foals. Fifty healthy (*n* = 15) and diseased (*n* = 35) Warmblood foals were examined. A left-sided and right-sided holosystolic murmur was audible in 98% (*n* = 42) and 51% (*n* = 22), respectively, on day 2; in 81% (*n* = 25) and 19% (*n* = 6) on day 5; and in 44% (*n* = 4) and 11% (*n* = 1) on day 10. The median grade of the systolic murmurs was higher when the DA was open. Flow through the DA could be visualized with color flow and continuous wave (CW) Doppler from the left parasternal long-axis view of the pulmonary artery in 40/43 foals on day 2, 9/31 foals on day 5 and 2/9 foals on day 10. The DA diameter was 2 ± 1 mm on day 2, 2 ± 1 mm on day 5 and 1 mm on day 10. The thickness of both septa of the FO was similar. The SP fluttered into the left atrium at all ages, but the maximal distance between the SP and SS decreased over time. In conclusion, cardiac murmurs, a patent DA and fluttering FO are frequent findings in neonatal foals. While these findings are probably physiological, the clinical importance needs to be further elucidated.

## 1. Introduction

In contrast to other species, the adaptation of the cardiac circulation to extra-uterine life is relatively unexplored in horses [1,2]. Both the ductus arteriosus (DA), also called the arterial duct, and the foramen ovale (FO) or oval foramen are important structures in fetal circulation [3]. During embryologic development, the primitive atrium is divided into a left and right atrium by the septum primum (SP) and septum secundum (SS). The FO is a tube-like passageway from the aperture of the vena cava, through the foramen secundum, which consists of fenestrations in the septum primum, to the lumen of the left atrium (LA) [4]. The SP forms the fluttering valve of the FO, which permits shunting of oxygen-rich placental blood via the interatrial septum from the right to the left heart and systemic circulation. After parturition, increased pressure in the LA causes progressive fusion of the SP and SS to close the atrial septum, with the fossa ovalis as a remnant of the FO [4,5].

During fetal life, deoxygenated blood also bypasses the lungs via the DA, which connects the pulmonary trunk to the descending aorta. The lumen of the DA is kept open due to low fetal systemic arterial oxygen and high circulating prostaglandin [6]. At birth, DA closure initially takes place by vasoconstriction due to increased arterial oxygen and decreased prostaglandins [1]. The timing and mechanism of DA closure in healthy and diseased foals have not yet been fully elucidated, and study results are conflicting. Scott et al. demonstrated closure at 16 h post-partum in healthy foals by cardiac catheterization and angiography [7]. Other studies suggest a histological and functional closure at 3 days post-partum [8,9,10]. A patent ductus arteriosus (PDA) has been defined as persistent flow through the DA beyond one week of age [11]. On the other hand, persistent echocardiographic turbulent flow through the DA could be visualized in 85% of normal pony foals at 35 days and in 43% at 49 days of age [12]. Doppler studies assessing DA closure are scarce in part due to the difficult echocardiographic diagnosis of patent DA because the lungs often hamper visualization of the descending aorta and DA [11,13]. To our knowledge, echocardiographic studies of the FO in neonatal foals are not available. The aim of this study is to describe the echocardiographic features of the DA and FO during the first 10 days of age in a hospital-based population of healthy and diseased foals.

## 2. Material and Methods

### 2.1. Animals

Cardiac ultrasound was prospectively performed on days 2, 5 and 10 after parturition in client-owned healthy and diseased neonatal Warmblood foals born at or admitted to the Department of Internal Medicine, Reproduction and Population Medicine, Ghent University. The study protocol was approved by the Ethics Committee on Animal Research and Testing of the Faculty of Veterinary Sciences (EC2021-009). Clinical data recorded included age, breed, gender, weight, gestational length, presence or absence of disease, heart rate, presence or absence of a cardiac murmur, type and duration of cardiac murmur, and therapy administered at the moment of the first ultrasound.

### 2.2. Echocardiography and Flow Doppler Examination

Two-dimensional echocardiographic images (VIVID IQ or VIVID E95, GE Healthcare, Diegem, Belgium) were acquired in lateral recumbency from the left and right hemithorax. Standardized right parasternal views included the four-chamber view (R-4C), left ventricular outflow tract view (R-LVOT), right ventricular inflow-outflow tract view (R-RVOT), short-axis view of the left (LV) and right ventricle (RV) at the chordal level (R-LVSAX_ch_) and short-axis view of the LA and aorta (Ao) (R-AoSAX) [14,15]. The FO was visualized on the R-4C view (Figure 1; Appendix A) and additionally using an oblique view which was obtained by rotating the ultrasound probe towards two o’clock and angulating dorso-caudally to visualize the FO and the caudal vena cava (Figure 2). All valves were evaluated for structural abnormalities. The DA could be visualized in the left parasternal long-axis view of the PA (L-PALAX) using a slight dorso-caudal probe angulation (Figure 3) [16] and from a slightly cranially angled R-RVOT view (Figure 4). Pulsed wave (PW) Doppler of pulmonary flow and continuous wave (CW) Doppler of DA flow was performed. Images were digitally stored, and offline analysis was performed using dedicated software (Echopac version BT203, GE Medical System, Diegem, Belgium). Color flow Doppler examinations were performed of all valves, the FO and the DA (Appendix A).

From the R-4C, the left atrial diameter was measured parallel to the mitral valve at end-diastole (R-4C LADd_end_) and end-systole (R-4C LADs_end_), and both left atrial and left the ventricular area were measured at end-diastole (R-4C LAAd_end_ and R-4C LVAd_end_) and end-systole (R-4C LAAs_end_ and R-4C LVAs_end_). The maximal SP and SS thickness was measured as well as the maximal distance between both. Left and right ventricular internal diameter were measured at the chordal level at end-diastole and peak systole (R-LVSAX_ch_ M-modeLVIDd_end_ and R-LVSAX_ch_ M-modeLVIDs_peak_, R-LVSAX_ch_ M-modeRVIDd_end_ and R-LVSAX_ch_ M-modeRVIDs_peak_) from the R-LVSAX M-mode. Pulmonary valve diameter was measured at the valvular level from the R-RVOT view at end-diastole (R-RVOT PADd_end_). The peak systolic and end-diastolic aortic root diameter at sinotubular level (R-LVOT AoDs_peak_ and R-LVOT AoDd_end_) and the pulmonary artery diameter at peak systole (R-LVOT PADs_peak_) were both measured in the R-LVOT view, and the pulmonary artery/aortic root ratio at peak systole (R-LVOT PADs_peak_/AoDs_peak_) was calculated. Pulmonary artery maximal flow velocity (PA Velmax) and velocity time integral (PA VTI; cm) were measured from the PW Doppler trace on the L-PALAX view. The DA was classified as open when turbulent flow could be visualized using color flow Doppler, and systolic flow through the DA could be measured using CW Doppler. Timing (systolic, diastolic) and maximal velocity (DA Velmax) was recorded on the CW Doppler flow profile. The DA internal diameter was measured as the maximal diameter of the ductus from the L-PALAX view (Figure 3).

### 2.3. Data Analysis

Statistical analysis was performed using SPSS version 27 (SPSS Inc., Chicago, IL, USA). The data distribution was evaluated by visual inspection, the Shapiro–Wilk test of linearity and the Kolmogorov–Smirnov test. Normally distributed data are reported as mean ± standard deviation, while non-normally distributed data are reported as median and range [minimum–maximum]. The cardiac murmur grade was compared between examinations with open versus closed DA using a Mann–Whitney U-test. The level of significance was *p* < 0.05.

## 3. Results

### 3.1. Study Population

Fifty warmblood foals, spread over the 3 examination days, met the inclusion criteria, of which 30% (*n* = 15) were healthy, and 70% (*n* = 35) were diseased (Table 1). Clinical and echocardiographic data were available on day 2 in 43 foals, on day 5 in 31 foals and on day 10 in 9 foals. Sixty percent (*n* = 30) were males and 40% (*n* = 19) were females. One foal’s gender was not recorded. The median weight was 55 kg (range of 26–77). The median gestational length was 330 days (range of 308–355 days). NSAIDs (Meloxicam 0.6 mg/kg IV or PO or Ketoprofen 2.2 mg/kg IV) were administered in 9 foals, intranasal humidified oxygen in 4 foals and natural colloids (plasma) or inotropes (Dobutamine 1–5 μg/kg/min) in 17 foals. Twenty-three foals only received other medication such as antibiotics. Fifteen healthy foals did not receive any medical treatment.

### 3.2. Cardiac Auscultation

On day 2, a left-sided holosystolic murmur grade 1–5/6 was audible in 98% of foals (*n* = 42/43) and a right-sided holosystolic murmur grade 1–4/6 in 51% (*n* = 22/43). A left-sided early diastolic murmur grade 1–3/6 could be detected in 14% of the foals (*n* = 6/43). All foals with a diastolic murmur also had a systolic murmur. On day 5, a left-sided holosystolic murmur grade 1–4/6 was audible in 81% (*n* = 25/31) and a right-sided holosystolic murmur grade 1–4/6 in 19% (*n* = 6/31). No diastolic murmurs were detected. On day 10, a left-sided holosystolic murmur grade 1–4/6 was audible in 44% (*n* = 4/9) and a right-sided holosystolic murmur grade 1–4/6 in 11% (*n* = 1/9). No diastolic murmur was audible. A left-sided systolic murmur grade 4/6 or more was detected in 43% (*n* = 19/43) of foals on day 2, in 29% (*n* = 9/31) on day 5 and in 11% (*n* = 1/9) on day 10.

The median grade of the left-sided systolic murmur was higher when the DA was open (3/6, range 0–5/6) compared to exams in which the DA was closed (2/6, range of 0–4/6, *p* < 0.001). The median grade of the right-sided systolic murmur was also higher in exams in which the DA was open (0/6, range 0–4/6) compared to closed (0/6, range of 0–3/6, *p* = 0.007). There was no significant difference in the degree of diastolic murmurs.

### 3.3. Echocardiographic Features

The echocardiographic measurements are reported in Table 2 and Table 3. The images of two foals were excluded from the analysis: one foal due to the concurrent presence of congenital pulmonary valve malformation and another due to inadequate image quality.

Turbulent flow through the DA could be visualized with color flow Doppler and CW Doppler from the L-PALAX view in 40/43 foals on day 2, in 19/31 on day 5 and in 2/9 foals on day 10. In all cases, the shunt was unidirectional from the left to the right side. In one foal on day 2, it was not possible to image the area of the DA due to lung artifact. The diastolic component could only be identified on the CW Doppler profile in 14/40, 5/19 and 0/2 foals on days 2, 5 and 10, respectively (Table 3). Color flow Doppler through the DA could be visualized on the slightly angled R-RVOT view in 23/40 foals on day 2, in 11/19 on day 5 and in none on day 10. The DA could be clearly visualized for two-dimensional measurement of the internal diameter from an L-PALAX view in 19/40 foals on day 2, 5/19 on day 5 and 1/2 on day 10 (Table 3). The diameter was 2 ± 1 mm on day 2, 2 ± 1 mm on day 5 and 1 mm on day 10.

The FO is a cone-shaped structure protruding into the LA. The SP showed a fluttering movement into the LA throughout the cardiac cycle in all foals at all ages. In some, the SP was so large that it filled a large part of the LA and could be seen at the ostium of pulmonary vein III. This could be visualized on the R-4C view, focused on the FO (Appendix A). The maximal distance between SP and SS decreased over time. The distance between both the SP and SS was at its largest during diastole. The thickness of both septa was similar (Table 3). Although color flow Doppler indicated flow between the SP and SS in all foals, none of them showed a visible jet of flow through the FO.

In the 10 foals in which NSAIDs had been administered, the DA was open in eight foals on day 2; in one, it was closed, and in one, image quality was not sufficient. In five out of eight, the DA was closed at 5 days of age; in one out of three, the ultrasound was repeated at 10 days of age, and the DA was closed.

Of the 43 foals examined on day 2, eight foals were premature (gestational length < 320 days; median 313 (range of 308–317) days), and in all, the DA was open. On day 5, an ultrasound was performed in seven out of eight premature foals, and in five out of seven, the DA was still open. In three of the foals with an open DA, a repeat exam was performed, which showed that the DA was now closed.

## 4. Discussion

This study describes auscultation and echocardiographic findings in healthy and diseased Warmblood foals. The presence of a cardiac murmur, a patent DA and a fluttering SP are frequent findings in neonatal foals of 2, 5 and 10 days of age.

In equids, several isolated and complex congenital cardiac defects have been reported. These are usually diagnosed after a cardiac murmur has been detected [13,17]. However, loud cardiac murmurs in foals shortly after birth in the absence of signs of cardiovascular compromise can be physiological findings. Those physiological, mostly systolic murmurs can be generated by turbulence in the LVOT, by turbulence in the DA or in the ductus diverticulum, which is the remaining cavity in the PA associated with a functionally closed DA [18]. In our study, nineteen foals had a left-sided systolic murmur grade 4/6 or more on day 2. This is in line with previous studies in which systolic murmurs were audible the first days of life [18,19], and in one study, 32% of the foals still had an audible systolic murmur at 16 weeks of age [2]. We suggest those audible murmurs are due to the etiologies cited above. In our study, the left- and right-sided systolic murmurs were statistically significantly louder in foals with a patent DA compared to foals with a closed DA. However, this difference may not be clinically relevant, as grade 0–4/6 left-sided systolic murmurs could be detected both in foals with a closed DA and foals with a patent DA on echocardiography.

In foals with cardiac murmurs and clinical signs related to the cardiovascular system, the echocardiographic assessment involves a systematic evaluation of each cardiac structure individually to determine its structure and its position in relation to other structures [20]. The cardiac dimensions in our study were smaller compared to the study of Collins et al. in thoroughbred foals [2]. This might be explained by the different breeds and lower body weights of the foals in our study. Both the DA and the FO could be visualized in two-dimensional right and left parasternal images. Turbulent flow through the DA was more often visualized on the L-PALAX view compared to the slightly angled RVOT view. This can be explained by the more superficial location of the DA and better flow alignment when echocardiography is performed from the left side. To visualize the DA in the L-PALAX view, the ultrasound beam must be angled dorsally and slightly caudally. Imaging this structure can be difficult or even impossible due to reverberations caused by the lung [9,11,13,21,22]. In our study, the velocity of flow through the DA measured by CW Doppler echocardiography seemed to decrease over the first 10 days of life. This was a surprising finding as the diameter of the DA is expected to decrease, which should result in similar or higher flow velocities. This might be related to the fact that different foals were included in each age group. Other potential explanations may be differences in pressure between the PA and the Ao, malalignment of the CW cursor or due to differences in DA morphology as described in dogs [23]. The physiological mechanisms and timing of DA closure are not entirely known in horses, and echocardiographic studies are scarce [13]. In our study, the DA was still open at 5 and even 10 days of age in some healthy and diseased foals. This contrasts with the study by Machida et al., which demonstrated morphometric and histological ductal closure at necropsy in nine foals aged 3–11 days [7]. This might be explained by the differences in diagnostic techniques, with our study using high-end ultrasound machines with sensitive Doppler measurements.

The clinical significance of an open DA at 10 days of age remains undetermined. Gestation length, systemic or respiratory disease and administration of medication may influence DA closure. Perinatal hypoxia may delay or reverse ductal closure, leading to persistent fetal circulation [24]. In premature animals, the DA often fails to close as the premature ductus is less likely to constrict, easily reopens due to insensitivity to elevated arterial oxygen tension and loses its ability to respond to vasoconstrictive stimulants with increasing postnatal age [6]. The premature ductus is also more sensitive to the effects of prostaglandins that are produced with inflammation, which contributes to the reopening of the ductus. Structural anatomic closure is also less likely to occur due to the larger diameter of the premature DA compared to a full-term DA [6]. In our study, the number of premature foals was low, and the relationship between DA closure and gestational length could not be evaluated. Similarly, differences in DA closure between healthy and diseased foals were difficult to assess due to confounding factors such as prematurity, perinatal hypoxia, and the administration of intranasal oxygen and medication. In humans, indomethacin (COX inhibitor) has been used for the closure of patent DA by lowering prostaglandin levels [6]. Administration of NSAIDs might also result in earlier DA closure. This did not seem to be the case in our study population, although statistical tests were not performed. Other treatment options for a patent DA in human and companion animal medicine include surgical ligation and minimally invasive occlusion with an occluder device [3]. In adult horses, a patent DA is uncommon as an isolated defect, and patent DA is often part of complex congenital cardiac defects [1,3,9,13,21]. A patent DA usually does not cause clinical signs during the first few months of life [13]. If the shunt is large, progressive left heart volume overload and ultimately left-sided congestive heart failure can develop [13]). Continuous machinery murmurs have been described in horses with a patent DA [11], but the sensitivity of the presence of a murmur for detecting a patent DA is only 40% [12].

Compared with previous studies [2], this study is the first to describe in detail the echocardiographic features of the FO in foals. In equine fetuses, the FO is a cone-shaped structure protruding into the LA, and in some, the SP is so large that it fills a large part of the LA and can be seen at the ostium of pulmonary vein III [16]. In neonatal foals post-mortem, the tubular structure of the FO appeared to have collapsed against the atrial wall, with the non-fenestrated portion against the right atrium [4]. Scanning electron microscopy demonstrated a knotted appearance of the collapsed tissue, which was suggested to be associated with active muscular contraction contributing to functional closure of the FO immediately after birth. However, our study demonstrated fluttering of the tube-like flap of the SP towards the LA until 10 days after birth. Flow has been demonstrated through the FO in some foals for several weeks after birth [13]. This was not detected in our study. However, turbulent flow between the two folds of the FO might be present in the absence of an interatrial shunt and could be misinterpreted as a patent FO. On the other hand, FO flow velocities might be very low and difficult to detect on transthoracic ultrasound. Intracardiac ultrasound might be useful to obtain more detail on the FO morphology and flow [25].

This study has several limitations. The study population was a small and hospital-based warmblood population. Therefore, echocardiographic examinations could only be performed when the foals were hospitalized, and several foals were discharged from the clinic or died secondary to problems unrelated to the cardiovascular system before the exam on day 5 or day 10 could be performed. The classification of open or closed DA was based on color flow and CW Doppler, as no gold standard examination is available. Ultrasound of the DA proved to be more difficult in some foals. Although the authors believe that suitable images were obtained in foals, the presence of a patent DA might have been underestimated due to image quality.

## 5. Conclusions

Cardiac murmurs, a patent DA and a fluttering FO are frequent findings in healthy and diseased neonatal foals. Those findings are probably physiological, although the clinical importance needs to be further elucidated. Longitudinal follow-up studies in healthy and diseased foals beyond 10 days of age are needed to exactly assess the time of closure of both the DA and the FO.

## Figures and Tables

**Figure 1 animals-12-02242-f001:**
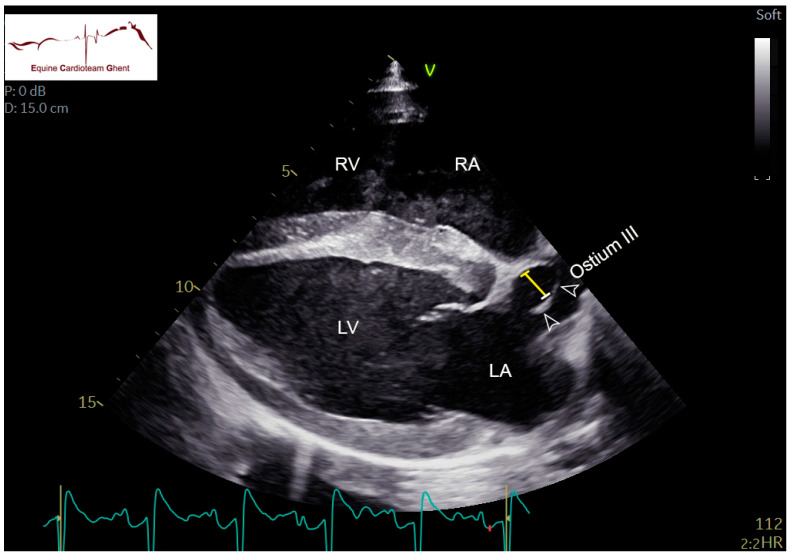
Right parasternal four-chamber (R-4C) view of the foramen ovale in a 2-day-old healthy foal. The arrowheads indicate the fluttering septum primum at the base of the ostium of pulmonary vein III (ostium III). The yellow bar indicates a 1 cm distance between the septum primum and septum secundum. LA: left atrium; LV: left ventricle; RA: right atrium; RV: right ventricle.

**Figure 2 animals-12-02242-f002:**
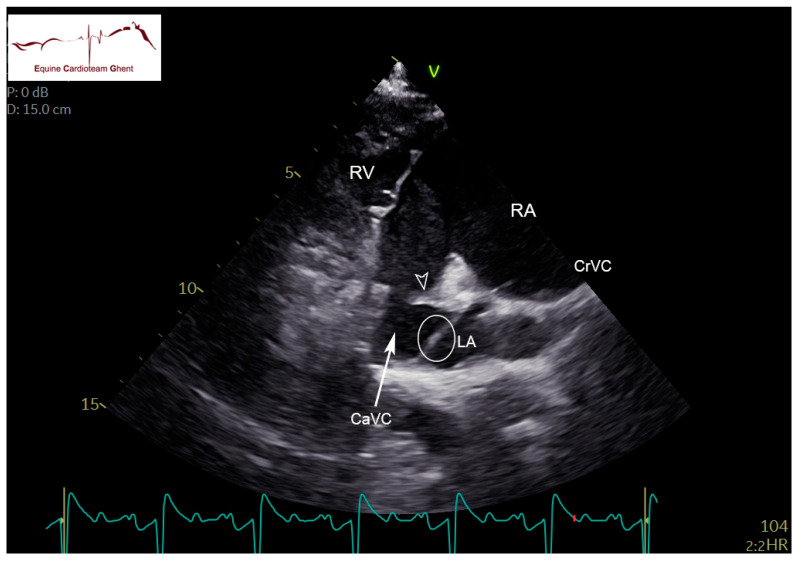
Septum primum (**circle**) and limbus (**arrowhead**) in a 2-day-old healthy foal visualized on a right parasternal oblique view obtained by rotating the ultrasound probe towards two o’clock and angulating dorso-caudally). CaVC: cranial vena cava, CrVC: cranial vena cava, LA: left atrium; RA: right atrium; RV: right ventricle.

**Figure 3 animals-12-02242-f003:**
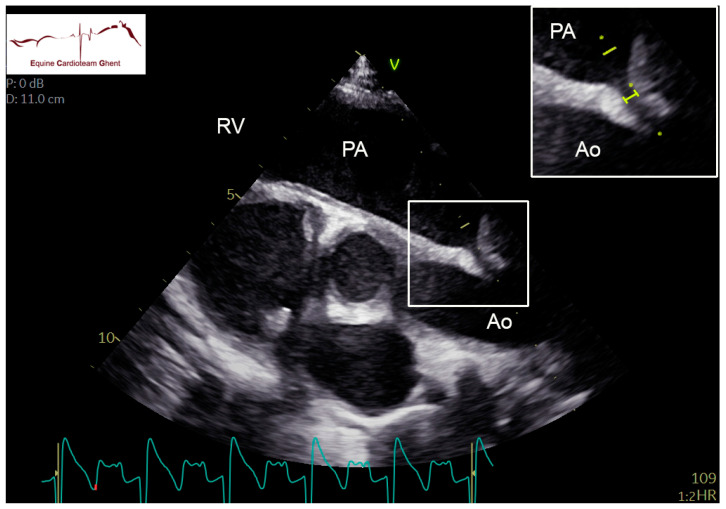
Left parasternal long-axis view of the pulmonary artery (PA) using slight dorso-caudal probe angulation in a 2-day-old healthy foal. The internal diameter of the ductus arteriosus is indicated by the yellow bar on the picture insert. The yellow dots indicate the cursos placement for continuous wave Doppler examination. Ao: aorta; PA: pulmonary artery; RV: right ventricle.

**Figure 4 animals-12-02242-f004:**
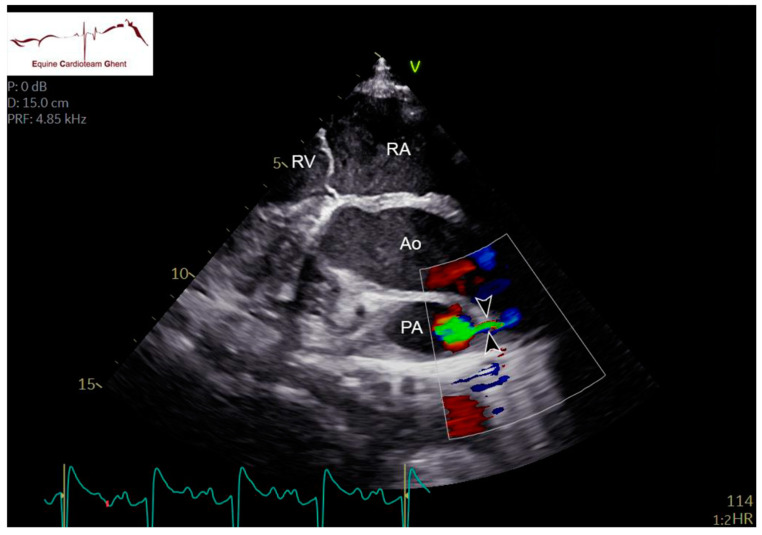
Color flow Doppler of the ductus arteriosus (**arrowheads**) in a 2-day-old healthy foal on the right parasternal slightly cranially angled R-RVOT view. Ao: aorta; PA: pulmonary artery; RA: right atrium; RV: right ventricle.

**Table 1 animals-12-02242-t001:** Number of foals at the different time points of the echocardiographic exam, with their initial clinical diagnosis; (ARDS: acute respiratory distress syndrome).

Initial Diagnosis	Day 2	Day 5	Day 10
	(*n* = 43)	(*n* = 31)	(*n* = 9)
Healthy foals	15	7	2
Premature foals (gestation length < 320 days)	8	7	3
Neonatal hypoxic encephalopathy, perinatal asphyxia syndrome, septicemia, neonatal isoerythrolysis, failure of passive transfer, foals born by cesarean section, gastrointestinal (colic, inguinal hernias, meconium impaction) and/or respiratory (ARDS) diseases	17	15	4
Orthopedic abnormalities (angular and flexural limb deformities)	3	2	0

**Table 2 animals-12-02242-t002:** Echocardiographic visualization of the ductus arteriosus (DA) from the left parasternal long-axis view of the pulmonary artery in healthy and diseased foals on days 2, 5 and 10. CW: continuous wave Doppler.

	Day 2		Day 5		Day 10	
	(*n* Open/*n* Total)		(*n* Open/*n* Total)		(*n* Open/*n* Total)	
	Healthy	Diseased	Healthy	Diseased	Healthy	Diseased
Turbulent flow through DA (Color Doppler)	13/15	27/28	6/7	13/24	0/2	2/7
Systolic flow through DA	13/13	27/27	6/6	13/13	-	2/2
(CW Doppler)						
Diastolic flow through DA	5/13	9/27	2/6	4/13	-	0/2
(CW Doppler)						

**Table 3 animals-12-02242-t003:** Echocardiographic measurements in 48 healthy and diseased foals at 2, 5 and 10 days of age.

	Day 2 (*n* = 43)	Day 5 (*n* = 31)	Day 10 (*n* = 9)
	Mean ± sd	Mean ± sd	Mean ± sd
HR (beats per minute)	102 ± 16	96 ± 18	93 ± 29
R-4C LADd_end_ (cm)	3.8 ± 0.5	3.8 ± 0.7	3.7 ± 0.6
R-4C LADs_end_ (cm)	4.2 ± 0.6	4.1 ± 0.7	4.2 ± 0.8
R-4C LAAd_end_ (cm^2^)	9.5 ± 2.2	10 ± 3.6	8.7 ± 2.6
R-4C LAAs_end_ (cm^2^)	16 ± 3	17 ± 4.2	13.8 ± 4.0
R-4C LVAd_end_ (cm^2^)	31 ± 6.1	36 ± 9.2	38.5 ± 9.0
R-4C LVAs_end_ (cm^2^)	16 ± 4.4	17 ± 6.6	19.7 ± 3.7
R-LVSAX_ch_ M-modeLVIDd_end_ (cm)	5.1 ± 0.7	5.3 ± 0.7	5.3 ± 1.0
R-LVSAX_ch_ M-modeLVIDs_peak_ (cm)	3.1 ± 0.6	3.4 ± 0.6	3.6 ± 0.8
R-LVSAX_ch_ M-modeRVIDd_end_ (cm)	2.1 ± 0.6	2.2 ± 0.6	2.2 ± 1.1
R-LVSAX_ch_ M-modeRVIDs_peak_ (cm)	1.3 ± 0.5	1.3 ± 0.6	1.2 ± 0.8
R-RVOT PADd_end_ (cm)	2.3 ± 0.2	2.2 ± 0.4	2.2 ± 0.5
R-LVOT PADs_peak_ (cm)	1.8 ± 0.3	1.9 ± 0.3	1.9 ± 0.2
R-LVOT AoDs_peak_ (cm)	2.6 ± 0.3	2.7 ± 0.3	2.7 ± 0.5
R-LVOT AoDd_end_ (cm)	2.2 ± 0.3	2.4 ± 0.4	2.3 ± 0.5
R-LVOT PADs_peak_/AoDs_peak_	0.7 ± 0.1	0.7 ± 0.1	0.7 ± 0.2
L-PALAX PWD PA Velmax (m/s)	1.2 ± 0.3	1.2 ± 0.3	1.1 ± 0.4
L-PALAX PWD PA VTI (cm)	24 ± 8.0	23 ± 5.0	22 ± 5.9
L-PALAX DAD (mm) *	2 ± 1	2 ± 1	1
L-PALAX CWD DA Velmax (m/s)	3.8 ± 0.6	3.3 ± 1.0	2.7 ± 1.1
R-4C SP thickness (mm)	3 ± 1	4 ± 1	4 ± 2
R-4C SS thickness (mm)	3 ± 1	3 ± 1	4 ± 2
Maximal distance between SP and SS (mm)	12 ± 5	11 ± 4	6 ± 2

AoDd_end_: aortic diameter at end diastole; AoDs_peak_: aortic diameter at peak systole; CWD: continuous wave Doppler; DA: ductus arteriosus; DAD: ductus arteriosus diameter; HR: heart rate; LAAd_end_: left atrial area at end diastole; LAAs_end_: left atrial area at end systole; LADd_end_: left atrial diameter at end diastole; LADs_end_: left atrial diameter at end systole; L-PALAX: left-sided pulmonary artery long axis view; LVAd_end_: left ventricle area at end diastole; LVAs_end_: left ventricle area at end systole; M-modeLVIDdend: M-mode left ventricle internal diameter at end diastole; M-modeLVIDs_peak_: M-mode left ventricle internal diameter at peak systole; M-modeRVIDd_end_: M-mode right ventricle internal diameter at end diastole; M-modeRVIDs_peak_: M-mode right ventricle internal diameter at peak systole; PA: pulmonary artery; PADd_end_: pulmonary artery diameter at end diastole; PADs_peak_: pulmonary artery at peak systole; PWD: pulsed wave Doppler; R-4C: right four-chamber view; R-LVOT: right-sided left ventricle outflow tract; R-LVSAX_ch_: right-sided left ventricle short axis chordae level; R-RVOT: right-sided right ventricle outflow tract; SP: septum primum; SS: septum secundum; Velmax: maximal velocity; VTI: velocity time integral. * day 2 *n* = 19, day 5 *n* = 5 and day 10 *n* = 1.

## Data Availability

The data presented in this study are available on request from the corresponding author.

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
