# Peer review of "Echocardiographic Features of the Ductus Arteriosus and the Foramen Ovale in a Hospital-Based Population of Neonatal Foals"

_animals, 2022, doi:10.3390/ani12172242_

Round 1
Reviewer 1 Report
I think this is a really interesting and clinically relevant subject for equine cardiology. The manuscript is generally well written and the images are excellent.
It would be nice if some sections of the discussion related better to your actual results. Some sentences are just statements and could be integrated better. Some statements are not backed up with the data you present in the study. Some of your results could be explained further in the discussion. I have included some specifics below.
Line 96. This is repeating line 93. Can 'and from a slightly cranially angled R-RVOT view' not just be added onto Lines 93-94?
Line 101. Where was the diameter of the LA measured. Widest diameter?
Line 124. Would 'distance' be a better word than 'thickness'
Fig 2. It is not clear what the arrows of CaVC and CrVC are indicating. Is blood not coming from CrVC to RA?
Fig 2. Line 128. The FO is circled. The FO is an opening in the SS, an opening is not visualised here though. Is this the SP?
Fig 2. Line 131. The arrowhead indicates the interatrial septum. Do you mean the SS?
Fig 2. Is that a foramen in the septum that the CaVC arrow is highlighting? In the summary, line 19, it says the FO was not open.
Line 136. PV is in the legend but not on the actual figure.
Line 140. TV is in the legend but not on the actual figure.
Lines 170-171. It would be nice to include the (n = ) values for these percentages, so the reader doesn't have to calculate it themselves.
Line 176. Whilst there may be statistical significance, is this magnitude of difference significant?
Line 190. Abbreviate to L-PALAX to be consistent
You mention in the summary that the FO was not open (line 19), but this isn't mentioned in the results.
Line 199. Is Fig 1 representative of the thicknesses of both septa? The SP seems quite a bit thicker than the SS in this image.
Table 2. Need to close the bracket of (CW Doppler) for systolic flow.
Line 235-236. Would this line not be better placed later when discussing examining foals beyond 10 days.
Line 240-242. This is just a statement. Can it be related to your work.
Line 242-248. I am not sure this is relevant. These are just statements not relevant to the results you have.
Line 256-257. 'The diagnosis of a patent DA is based ....' Do you have a ref for this or are you saying in your study it WAS based. This sentence is also repetitive of what you stated in your methods.
Line 257-258. I don't think you have any stats to back this statement up? You also have different foals in the age groups, some added and some dropped out, so you are not just longitudinally following an entire group.
Line 262 to 269. How is this related to the current study?
Line 271-272. Perhaps clarify in SOME foals. Only 2/9
Line 274. Would you not expect histology to be more sensitive than Doppler?
Line 286. There is no mention in the methods, what stats you did to show this?
Line 289. There is no mention in the methods, what stats you did to show this?
Would it be worth discussing that the L-PALAX view appeared to be more sensitive for DA than the R-RVOT and why?
Would it be worth discussing the differences between the healthy and diseased foals in your study?
Author Response
Dear reviewer,
Thank you for reviewing my article. Please find the answers and the changes below. Please find the changes made to the manuscript using track changes. The lines cited below are the line numbers when track changes is on “All markup”.
Line 96. This is repeating line 93. Can 'and from a slightly cranially angled R-RVOT view' not just be added onto Lines 93-94?
Yes, I agree. I adapted line 93. “The DA could be visualized in the left parasternal long-axis view of the PA (L-PALAX) using a slight dorso-caudal probe angulation (Fig. 3) [15] and from a slightly cranially angled R-RVOT view (Fig. 4).”
Line 101. Where was the diameter of the LA measured. Widest diameter?
Indeed, the widest diameter. This has been adapted and is now line 103 “From the R-4C, the maximal left atrial diameter was measured parallel to the mitral valve annulus at end-diastole”
Line 124. Would 'distance' be a better word than 'thickness'
The figure legends have been adapted for clarification.
Fig 2. It is not clear what the arrows of CaVC and CrVC are indicating. Is blood not coming from CrVC to RA? Fig 2. Line 128. The FO is circled. The FO is an opening in the SS, an opening is not visualised here though. Is this the SP? Fig 2. Line 131. The arrowhead indicates the interatrial septum. Do you mean the SS? Fig 2. Is that a foramen in the septum that the CaVC arrow is highlighting? In the summary, line 19, it says the FO was not open.
The figure annotations and figure legend have been adapted for clarification.
Line 136. PV is in the legend but not on the actual figure.
Yes indeed, I deleted it (it is now line 141).
Line 140. TV is in the legend but not on the actual figure.
Yes indeed, I deleted it (it is now line 146).
Lines 170-171. It would be nice to include the (n = ) values for these percentages, so the reader doesn't have to calculate it themselves.
This has been adapted and is now line 178 “A left-sided systolic murmur grade 4/6 or more was detected in 43% (n=19/43) of foals on day 2, in 30% (n=9/31) on day 5 and in 11% (n=1/9) on day 10.”
Line 176. Whilst there may be statistical significance, is this magnitude of difference significant?
We have added this to the discussion (line 252): “In our study, the left and right sided systolic murmurs were statistically significantly louder in foals with a patent DA compared to foals with a closed DA. However, this difference may not be clinically relevant, as grade 0-4/6 left-sided systolic murmurs could be detected both in foals with a closed DA and foals with a patent DA on echocardiography”.
Line 190. Abbreviate to L-PALAX to be consistent
Indeed, I adapted this (now line 198)
You mention in the summary that the FO was not open (line 19), but this isn't mentioned in the results.
Indeed, I added more information concerning this. Now in line 207 “Although color flow Doppler indicated flow in between the SP and SS in all foals, none of them showed a visible jet of flow through the FO.”
Line 199. Is Fig 1 representative of the thicknesses of both septa? The SP seems quite a bit thicker than the SS in this image.
This image was taken to optimally visualize the fluttering septum primum at the base of ostium III and was not an optimal image to measure the thickness of the SS and SP.
Table 2. Need to close the bracket of (CW Doppler) for systolic flow.
Indeed, has been adapted.
Line 235-236. Would this line not be better placed later when discussing examining foals beyond 10 days.
We chose to structure the discussion based on the findings, starting with the discussion of murmurs in foals and continuing with echocardiography. Therefore we opted to leave this line in this paragraph.
Line 240-242. This is just a statement. Can it be related to your work. Line 242-248. I am not sure this is relevant. These are just statements not relevant to the results you have.
These statements have been deleted.
Line 256-257. 'The diagnosis of a patent DA is based ....' Do you have a ref for this or are you saying in your study it WAS based. This sentence is also repetitive of what you stated in your methods.
Indeed. I adapted this to now line 271: “In our study the diagnosis of a patent DA was based on the presence of continuous flow visualized by color flow and CW Doppler echocardiography. Over the first 10 days of life the velocity through the DA decreased.”
Line 257-258. I don't think you have any stats to back this statement up? You also have different foals in the age groups, some added and some dropped out, so you are not just longitudinally following an entire group.
See answer above.
Line 262 to 269. How is this related to the current study?
The sentences have been rephrased in line 271 “In our study, the velocity of flow through the DA measured by CW Doppler echocardiography seemed to decrease over the first 10 days of life. This was a surprising finding as the diameter of the DA is expected to decrease which should result in similar or higher flow velocities. This might be related to the fact that different foals were included in each age group. Other potential explanations may be differences in pressure between the PA and the Ao, malalignment of the CW cursor or due to differences in DA morphology as described in dogs”
Line 271-272. Perhaps clarify in SOME foals. Only 2/9
Yes, I did clarify.
Line 274. Would you not expect histology to be more sensitive than Doppler?
Comparing both techniques is beyond the scope of our study but identifying a clear jet of flow from aorta to pulmonary artery, confirmed by CW Doppler, cannot be an artefact.
Line 286. There is no mention in the methods, what stats you did to show this?
This has been adapted and is now line 307 “In our study, the number of premature foals was low and the relationship between DA closure and gestational length could not be evaluated.”
Line 289. There is no mention in the methods, what stats you did to show this?
We have adapted this statement (now line 312): “Administration of NSAIDs might also result in earlier DA closure. This did not seem to be the case in our study population, although statistical tests were not performed.”
Would it be worth discussing that the L-PALAX view appeared to be more sensitive for DA than the R-RVOT and why?
Yes, this is worth be discussed. I added the following in line 271: “In our study, turbulent flow through the DA was more often visualized on the L-PALAX view compared to the slightly angled RVOT view This can be explained by the more superficial location of the DA and better flow alignment when echocardiography is performed from the left side”
Would it be worth discussing the differences between the healthy and diseased foals in your study?
This is an important comment however we preferred not to discuss those differences more in detail because of the confounding factors. The sample (numbers of foals, ages, dropouts) is not representative enough to compare healthy and diseased foals.

Reviewer 2 Report
The authors are commended on a clear, concise, well written descriptive study. The tables, figures and supplemental videos enhance the manuscript greatly and are well labeled and clearly described. This manuscript will prove to be an excellent resource for clinicians practicing equine neonatology.
Minor comments only.
Specific comments:
In regard to figure 1: Line 123: change "indicate" to indicates
Line 124: "thickness" is used to describe the distance between the ostium primum and ostium secondum in figure 1 and that imples there is a solid structure between the two membranes. My admittedly rudimentary understanding is that there is a potential space between these two thin membranous structures, in this case using distance, might be more appropriate. As I read further, It seems as though distance was used to describe this measurement in the body of the manuscript, so thickness is likely a typo.
Line 205-207: Are the authors reporting whether the DA was open or closed in premature foals? The numbers just make that a bit confusing. It might be helpful to say premature foals, instead of foals here. In line 207, is it possible that 3/5 is a typo? do the authors mean 3/8 premature foals had an exam on day 10 and all had closed DA?
Table 2: need to add a closed parentheses for CW doppler in the second column of the table
Table 3: please provide units for R-LVOT AoDspeak and R-LVOT AoDdend (I believe these should also be in cm)
Discussion: Line 307: please change up till to until
Line 320: Would it be more correct to say such that visualization of the DA was impeded or impossible in some instances (rather than challenging which implies you could still image it although it was difficult to do).
Author Response
Dear reviewer,
Thank you for reviewing my article. Please find the answers and the changes below (written in red). Please find the changes made to the manuscript using track changes. The lines cited below are the line numbers when track changes is on “All markup”.
Reviewer 2
The authors are commended on a clear, concise, well written descriptive study. The tables, figures and supplemental videos enhance the manuscript greatly and are well labeled and clearly described. This manuscript will prove to be an excellent resource for clinicians practicing equine neonatology.
Minor comments only.
Specific comments:
In regard to figure 1: Line 123: change "indicate" to indicates
This has been adapted.
Line 124: "thickness" is used to describe the distance between the ostium primum and ostium secondum in figure 1 and that imples there is a solid structure between the two membranes. My admittedly rudimentary understanding is that there is a potential space between these two thin membranous structures, in this case using distance, might be more appropriate. As I read further, It seems as though distance was used to describe this measurement in the body of the manuscript, so thickness is likely a typo.
The figure legends have been adapted for clarification.
Line 205-207: Are the authors reporting whether the DA was open or closed in premature foals? The numbers just make that a bit confusing. It might be helpful to say premature foals, instead of foals here. In line 207, is it possible that 3/5 is a typo? do the authors mean 3/8 premature foals had an exam on day 10 and all had closed DA?
I added “premature” and rephrased in order to make the text clearer.
Table 2: need to add a closed parentheses for CW doppler in the second column of the table
Yes, indeed, this has been adapted.
Table 3: please provide units for R-LVOT AoDspeak and R-LVOT AoDdend (I believe these should also be in cm)
This has been adapted.
Discussion: Line 307: please change up till to until
This has been adapted.
Line 320: Would it be more correct to say such that visualization of the DA was impeded or impossible in some instances (rather than challenging which implies you could still image it although it was difficult to do).
Visualization of the DA region was more difficult in some foals but we do believe that we could obtain sufficient detail in all foals. However, we cannot be entirely sure as there was no autopsy. We have rephrased the sentence.

Reviewer 3 Report
Detailed remarks are contained in the attached file. The most notable concerns are the omission of one relevant reference and many incomplete citations in the references list.

Author Response
Reviewer 3
Detailed remarks are contained in the attached file. The most notable concerns are the omission of one relevant reference and many incomplete citations in the references list.
All adaptations and clarifications requested by the reviewer were performed. The citations in the references list were adapted.
Key reference missing: Collins, Palmer and Marr. Two-dimensional and M-mode echocardiographic findings in healthy Thoroughbred foal. (AVJ 2010)
This reference was included in the manuscript as reference number 2. Adaptations were made were requested. Following line 267 was added “The cardiac dimensions in our study were smaller compared to the study of Collins et al. in Thoroughbred foals [2]. This might be explained by the different breed and lower body weight of the foals in our study.”
